# Generalization Gap in Amortized Inference

**Mingtian Zhang**          **Peter Hayes**          **David Barber**

Centre for Artificial Intelligence, University College London
{m.zhang,p.hayes,d.barber}@cs.ucl.ac.uk

## Abstract

The ability of likelihood-based probabilistic models to generalize to unseen data is central to many machine learning applications such as lossless compression. In this work, we study the generalization of a popular class of probabilistic model - the Variational Auto-Encoder (VAE). We discuss the two generalization gaps that affect VAEs and show that overfitting is usually dominated by amortized inference. Based on this observation, we propose a new training objective that improves the generalization of amortized inference. We demonstrate how our method can improve performance in the context of image modeling and lossless compression.

## 1   Introduction

Probabilistic models have achieved great success in many machine learning applications [5, 3]. Given a set of training data that are sampled from an underlying data distribution $\mathcal{X}_{train} = \{x_1, \cdots, x_N\} \sim p_d(x)$, the goal of probabilistic modelling is to approximate $p_d(x)$ with a model $p_\theta(x)$. A principled method to learn $\theta$ is to minimize the Kullback-Leibler (KL) divergence

$$\text{KL}(p_d(x)||p_\theta(x)) = \langle \log p_d(x) \rangle_{p_d(x)} - \langle \log p_\theta(x) \rangle_{p_d(x)}, \tag{1}$$

where we use $\langle \cdot \rangle$ to denote integration: $\langle f(x) \rangle_{p(x)} \equiv \int f(x)p(x)dx$. The first term represents the negative entropy of the data distribution $-H(p_d) \equiv \langle \log p_d(x) \rangle_{p_d(x)}$, which is a constant. The second cross entropy term involves the integration over the unkown data distribution $p_d(x)$, which can be approximated by the Monte-Carlo approximation using the training dataset $\mathcal{X}_{train}$

$$\langle \log p_\theta(x) \rangle_{p_d(x)} \approx \frac{1}{N} \sum_{n=1}^{N} \log p_\theta(x_n). \tag{2}$$

Therefore, estimating $\theta$ by minimizing the KL divergence is equivalent to Maximum Likelihood Estimation (MLE) when $N \to \infty$.

For a finite dataset, a common concern in both supervised and unsupervised learning is that the probabilistic model may overfit to the training dataset $\mathcal{X}_{train}$, degrading generalization performance [32]. The generalization performance in the unsupervised setting can be measured by the test likelihood [49]: $\frac{1}{M} \sum_{n=1}^{M} \log p_\theta(x'_m)$, where $\mathcal{X}_{test} = \{x'_1, \ldots, x'_M\} \sim p_d(x)$ is the test dataset. A model that has overfit to the training dataset $\mathcal{X}_{train}$ generally results in a high training likelihood but a low test likelihood. Although the test likelihood is a common evaluation criterion [36], the factors that affect the generalization of unsupervised probabilistic models are less well studied in comparison to supervised learning. We posit that this is because for common tasks, like sample generation or representation learning, good generalization in terms of the test likelihood is not a sufficient measure of performance. For example implicit models can generate sharp samples without having a likelihood function [16, 2, 46] and representations learned by latent variable models can be arbitrarily transformed without changing the likelihood [25]. However, in recent applications that use deep

generative models for lossless compression [38, 39, 22, 49, 47], generalization in terms of the test likelihood directly indicates higher compression rate [49]. Specifically, given a probabilistic model $p_\theta(x)$, a lossless compressor can be constructed to compress a test data point $x'$ to a bit string with length approximately equal to $-\log_2 p_\theta(x')$. When $p_\theta(x) \to p_d(x)$, the average compression length attains the entropy of the data distribution $-\frac{1}{M}\sum_{m=1}^{M}\log_2 p_\theta(x'_m) \to H(p_d)$, which is *optimal* under Shannon's source coding theorem [34], see Appendix E for a detailed introduction. Therefore, a better test likelihood can lead to a greater saving in bits and so understanding and improving generalization of deep generative models is an important challenge.

## 1.1 Variational Auto-Encoder

A popular type of probabilistic model is the Variational Auto-Encoder (VAE) [21, 29], which assumes a latent variable model $p_\theta(x) = \int p_\theta(x|z)p(z)dz$. For a nonlinear parameterization of $p_\theta(x|z)$ (e.g. a deep neural network), the evaluation of $\log p_\theta(x)$ involves solving an intractable integration over $z$. In this case, the evidence lower bound (ELBO) can be used to side-step the intractability

$$\langle\log p_\theta(x)\rangle_{p_d(x)} \geq \langle\log p_\theta(x,z) - \log q_\phi(z|x)\rangle_{q_\phi(z|x)p_d(x)} \equiv \langle\text{ELBO}(x,\theta,\phi)\rangle_{p_d(x)}, \quad (3)$$

where $q_\phi(z|x)$ is a variational posterior parameterized by a neural network with parameter $\phi$. The use of an approximate posterior of the form $q_\phi(z|x)$ is called *amortized inference*. To better understand this objective, we can rewrite the expected ELBO as the following

$$\langle\text{ELBO}(x,\theta,\phi)\rangle_{p_d(x)} = \left\langle\log p_\theta(x) - \text{KL}(q_\phi(z|x)||p_\theta(z|x))\right\rangle_{p_d(x)} \quad (4)$$

$$= \underbrace{-H(p_d)}_{const.} - \underbrace{\text{KL}(p_d(x))||p_\theta(x))}_{\text{model learning}} - \underbrace{\langle\text{KL}(q_\phi(z|x)||p_\theta(z|x))\rangle_{p_d(x)}}_{\text{amortized inference}}, \quad (5)$$

We denote the posterior family of $q_\phi(z|x)$ as $\mathcal{Q}$, which is indexed by a finite dimensional $\theta$ [43]. If $\mathcal{Q}$ is flexible enough such that the true posterior $p_\theta(z|x) \in \mathcal{Q}$, where $p_\theta(z|x) \propto p_\theta(x|z)p(z)$, then in the optimum of Equation 4, we have $\text{KL}(q_\phi(z|x)||p_\theta(z|x)) = 0 \Rightarrow q_\phi(z|x) = p_\theta(z|x)$ for $x \sim p_d(x)$ and the ELBO will be equal to the log-likelihood $\text{ELBO}(x,\theta,\phi) = \log p_\theta(x)$ [21, 6]. Many methods have been developed to increase the flexibility of $\mathcal{Q}$, e.g. adding auxiliary variables [1, 26] or flow-based methods [9, 28], to obtain a tighter ELBO.

Recent works [38, 39, 22] have successfully applied VAE style models to lossless compression realizing impressive performance. In this setting, the average compression length on the test data set is approximately equal to $-\frac{1}{M}\sum_{m=1}^{M}\text{ELBO}(x'_m,\theta,\phi)$ (also see Appendix E). Hence the better the test ELBO indicates the better the compression performance. This motivates us to study the factors that affect the generalization of VAEs and find practical ways to improve the generalization of VAEs.

The contributions of our paper are summarized as follows:

- We show the generalization of VAEs is affected by both the generative model (decoder) and the amortized inference network (encoder); and that the overfitting of VAEs is mainly dominated by the amortized inference.

- We propose a new training objective that can improve the generalization of the amortized inference without changing the model itself.

- We demonstrate how the proposed method can improve the compression rate in a practical lossless compression system without scarifying any computation speed.

## 2 Generalization of VAEs

During training, we only have access to a finite dataset $\mathcal{X}_{train}$, which leads to the following Monte-Carlo approximation as our objective to train VAEs:

$$\langle\text{ELBO}(x,\theta,\phi)\rangle_{p_d(x)} \approx \frac{1}{N}\sum_{n=1}^{N}\text{ELBO}(x_n,\theta,\phi). \quad (6)$$

This empirical approximation will lead to the VAE overfits to the training data for finite $N$. For example, we train a VAE on the Binary MNIST dataset for 1k epochs and plot the Bits-per-dimension (BPD)[1] of both training and testing dataset for every 100 epochs, also see Section 4 for model and training details. Figure 1 visualizes the training and testing BPD, which shows the VAE model is overfitting to the training dataset.

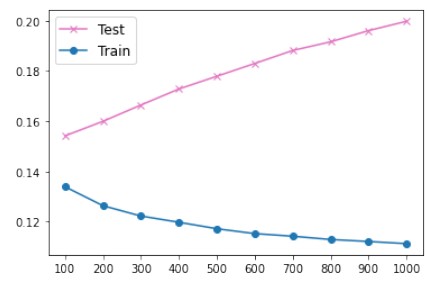

Figure 1: BPD vs epochs. The training BPD decreases but the testing BPD increases during training, which indicates the VAE overfits to $\mathcal{X}_{train}$.

The decomposition in Equation 5 suggests that the empirical ELBO contains 1) a *model empirical approximation*:

$$\mathrm{KL}(p_d(x))||p_\theta(x)) \approx \tfrac{1}{N}\sum_{n=1}^{N}\log p_\theta(x_n) + const., \quad (7)$$

which will potentially make a flexible model $p_\theta(x)$ overfit to the training data; and 2) an *amortized inference empirical approximation*:

$$\left\langle\mathrm{KL}(q_\phi(z|x)||p_\theta(z|x))\right\rangle_{p_d(x)} \approx \tfrac{1}{N}\sum_{n=1}^{N}\mathrm{KL}(q_\phi(z|x_n)||p_\theta(z|x_n)), \quad (8)$$

where similarly a flexible $q_\phi(z|x)$ can also overfit to the training data. More specifically, we let $\hat{\phi}$ be the optimal parameter of the empirical variational inference objective

$$\hat{\phi} = \arg\min_\phi \tfrac{1}{N}\sum_{n=1}^{N}\mathrm{KL}\left(q_\phi(z|x_n)||p_\theta(z|x_n)\right) \quad (9)$$

and we assume for any training data point $x_n \in \mathcal{X}_{train}$

$$q_{\hat{\phi}}(z|x_n) = \arg\min_{q\in\mathcal{Q}}\mathrm{KL}(q_\phi(z|x_n)||p_\theta(z|x_n)) \equiv q^*(z|x_n),$$

where $q^*(z|x_n)$ is the realizable optimal posterior (in the $\mathcal{Q}$ family) for $x_n$[2]. When $q_{\hat{\phi}}(z|x_n)$ overfits to $\mathcal{X}_{train}$, $q_{\hat{\phi}}(z|x'_m)$ may not be a good approximation to the true posterior $p_\theta(z|x'_m)$ for test data $x'_m \in \mathcal{X}_{test}$, We refer to the difference between the ELBO evaluated using $q_{\hat{\phi}}(z|x)$ and the ELBO evaluated using $q^*(z|x)$ as the *amortized inference generalization gap*, which is formally defined as

$$\left\langle\mathrm{KL}(q_{\hat{\phi}}(z|x)||p_\theta(z|x)) - \mathrm{KL}(q^*(z|x)||p_\theta(z|x))\right\rangle_{p_d(x)}. \quad (10)$$

Equivalently, this gap can be written as the difference between two ELBOs with two different $q$

$$\left\langle \underbrace{\left\langle\log p_\theta(x,z) - \log q^*(z|x)\right\rangle_{q^*(z|x)}}_{\text{ELBO with optimal inference}} - \underbrace{\left\langle\log p_\theta(x,z) - \log q_{\hat{\phi}}(z|x)\right\rangle_{q_{\hat{\phi}}(z|x)}}_{\text{ELBO with amortized inference}} \right\rangle_{p_d(x)}. \quad (11)$$

The inference neural network introduced by amortization is the cause of this inference generalization gap. It is important to emphasize that this gap cannot be reduced by simply using a more flexible $\mathcal{Q}$. This would only make $\mathrm{KL}(q_\phi(z|x_n)||p_\theta(z|x_n))$ smaller for the training data $x_n \in \mathcal{X}_{train}$ but would not explicitly encourage better generalization performance on test data [35].

To summarize, the generalization performance of a VAE depends on two factors:

- **Generative model generalization gap:** defined as $\mathrm{KL}(p_d(x)||p_\theta(x))$ and is caused by the generative model overfitting to the the training data.
- **Amortized inference generalization gap:** defined in Equation 11 and is caused by the amortized inference model (encoder) overfitting to the the training data.

## 2.1 Impact of the Generalization Gaps

The *generative model generalization gap* that is estimated by the test dataset (up to a constant) $\mathrm{KL}(p_d(x)||p_\theta(x)) \approx -\tfrac{1}{M}\sum_{m=1}^{M}\log p_\theta(x'_m) + const.$ cannot be calculated explicitly since we can

---

[1]In the case of VAE, the BPD is defined as the the negative ELBO (with a base 2 logarithm) normalized by the data dimension, lower BPD indicates higher ELBO.

[2]For a powerful inference network we assume that there is no amortization gap [11], which means $q_{\hat{\phi}}(z|x)$ can provide the optimal $q^*(z|x_n)$ for any training data $x_n \in \mathcal{X}_{train}$ - see Section 6 for further discussion.

only evaluate the lower bound $-\frac{1}{M}\sum_{m=1}^{M}\mathrm{ELBO}(x'_m,\theta,\phi)$. Fortunately, as suggested in Equation 4, if we know the optimal posterior for the test data $q^*(z|x'_m) \equiv \arg\min_{q\in\mathcal{Q}}\mathrm{KL}(q(z|x'_m)||p_\theta(z|x'_m))$, the log-likelihood can be approximated by the lower bound $\log p_\theta(x'_m) \approx \mathrm{ELBO}(x'_m,\theta,\phi)$ and the approximation becomes an equality when $p_\theta(z|x'_m) \in \mathcal{Q}$. Similarly, the *amortized inference generalization gap* can be estimated by knowing the optimal posterior $q^*(z|x'_m)$ for the test dataset:

$$\frac{1}{M}\sum_{m=1}^{M}\langle \log p_\theta(x'_m,z) - \log q^*(z|x'_m)\rangle_{q^*(z|x'_m)} - \langle \log p_\theta(x'_m,z) - \log q_{\hat\phi}(z|x'_m)\rangle_{q_{\hat\phi}(z|x'_m)}. \quad (12)$$

We can then estimate $q^*(z|x'_m)$ by fixing $\theta$ (which is trained on the training dataset) and learning $\phi^*$ on the test dataset and assuming $q^*(z|x'_m) = q_{\phi^*}(z|x'_m)$, where

$$\phi^* = \min_\phi \mathrm{KL}(q_\phi(z|x'_m)||p_\theta(z|x'_m)) = \max_\phi \langle \log p_\theta(x'_m,z) - \log q_\phi(z|x'_m)\rangle_{q_\phi(z|x'_m)}. \quad (13)$$

This optimal inference strategy can eliminate the effect of the inference generalization gap, allowing us to isolate the degree to which both the generative model and amortized inference generalization gaps are contributing to the overfitting.

We take the VAE described in Section 2 and train $q_\phi(z|x)$ for 1k epochs with Adam [20] and lr $= 5\times10^{-4}$ on the test data using Equation 13 to obtain the test BPD for the optimal inference strategy. In Figure 2 we plot the test ELBO (BPD) using the optimal inference strategy (green) and classic amortized inference (purple). Since for the optimal inference strategy the average likelihood $\frac{1}{M}\sum_{n=1}^{M}\log p_\theta(x'^{(m)})$ can be effectively approximated by the ELBO (see Appendix A for an empirical verification of the tightness of the ELBO), then the difference between the two inference curves on the test set (Test and Optimal) is the *amortized inference generalization gap*. We observe that after eliminating the inference generalization gap, the test BPD is stable with a marginal increase during training.

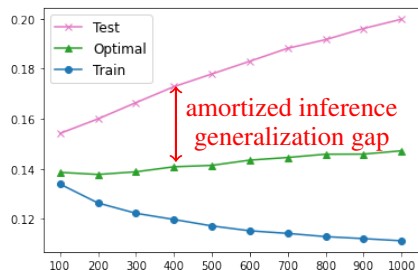

Figure 2: Test BDP vs epochs. Demonstrates the amortized inference generalization gap in a VAE trained on MNIST.

This suggests the generative model (decoder) slightly overfit to the data but that the overfitting is mainly dominated by the overfitting of the amortized inference network.

Although the optimal inference strategy can help eliminate the inference generalization gap, training $q_\phi$ on the test data is not practical in most applications of interest. Therefore, we now focus on improving the generalization of amortized inference without access to the test data at training time.

## 3  Improving Generalization with Consistent Amortized Inference

We now propose an *inference consistency requirement* which, if satisfied, would result in optimal generalization performance for amortized variational inference. Specifically when $p_\theta \to p_d$, the amortized posterior should converge to the true posterior $q_\phi(z|x) \to p_\theta(z|x)$[3] for every $x \sim p_d(x)$. Although this requirement seems natural for variational inference, the classic amortized inference training that is used for VAEs [21] doesn't satisfy it. Recall the typical VAE empirical ELBO training objective

$$\frac{1}{N}\sum_{n=1}^{N}\log p_\theta(x_n) - \mathrm{KL}(q_\phi(z|x_n)||p_\theta(z|x_n)). \quad (14)$$

When the model converges to the true distribution $p_{\theta^*} = p_d$ the training criterion for $q_\phi(z|x)$

$$\min_\phi -\frac{1}{N}\sum_{n=1}^{N}\mathrm{KL}(q_\phi(z|x_n)||p_{\theta^*}(z|x_n)) \quad (15)$$

can still result in the amortized posterior $q_\phi(z|x)$ overfitting to the training data. In principle, one could also limit the network capacity and/or add an explicit regularizer to the parameters [32] in an attempt to improve the generalization. However, this still cannot satisfy the consistency requirement in principle because it still only use the finite training dataset. Alternatively, there is another classic variational inference method that we now discuss, the wake-sleep training algorithm [13, 18], which does in fact satisfy the proposed consistency requirement.

---

[3]We assume the true posterior belongs to the variational family $p_\theta(z|x) \in \mathcal{Q}$.

## 3.1 Wake-Sleep Training

Defining $q_\phi(x, z) = q_\phi(z|x)p_d(x)$ and $p_\theta(x, z) = p_\theta(x|z)p(z)$, the two phases of the wake-sleep training [13, 18] can be written as minimizing two different KL divergences in both $x$ and $z$ space.

**Wake phase model learning:** $p_\theta(x|z)$ is trained by minimizing the KL divergence

$$\min_\theta \mathrm{KL}(q_\phi(x, z)||p_\theta(x, z)) = \max_\theta \left\langle \mathrm{ELBO}(x, \theta, \phi) \right\rangle_{p_d(x)} + const., \tag{16}$$

where $\langle \cdot \rangle_{p_d(x)}$ is approximated using the training set. This is referred to as the *wake phase* since the model is trained on experience from the 'real environment', i.e. it uses true data samples from $p_d(x)$.

**Sleep phase amortized inference:** $q_\phi(z|x)$ is trained by minimizing the KL divergence

$$\min_\phi \mathrm{KL}(p_\theta(x, z)||q_\phi(x, z)) = \min_\phi \left\langle \mathrm{KL}(p_\theta(z|x)||q_\phi(z|x)) \right\rangle_{p_\theta(x)} + const. \tag{17}$$

Leaving out the terms that are irrelevant to $\phi$, the objective can be estimated with Monte-Carlo $\langle - \log q_\phi(z|x'_m) \rangle_{p_\theta(x,z)} \approx \frac{1}{K} \sum_{k=1}^{K} - \log q_\phi(z_k|x_k)$, where $z_k \sim p(z)$ and $x_k \sim p_\theta(x|z_k)$. This is referred to as the *sleep phase* because the samples from the model used to train $q_\phi$ are interpreted as dreamed experience. In contrast, the training criterion for the typical VAE amortized inference (Equation 8) uses the true data samples from $p_d$ to train $q_\phi(z|x)$, which we refer to as *wake phase amortized inference*. We notice that if a perfect model $p_{\theta*}(x) = p_d(x)$ is used in the sleep phase amortized inference, then it is equivalent to minimizing

$$\left\langle \mathrm{KL}(p_\theta(z|x)||q_\phi(z|x)) \right\rangle_{p_{\theta*}(x)} = \left\langle \mathrm{KL}(p_\theta(z|x)||q_\phi(z|x)) \right\rangle_{p_d(x)}. \tag{18}$$

Therefore, the training of the inference network satisfies the *inference consistency requirement* since we can access infinite training data from $p_d$ by sampling from $p_{\theta*}$.

However, the wake-sleep algorithm presented lacks convergence guarantees [13] and minimizing $\mathrm{KL}(p_\theta(z|x)||q_\phi(z|x))$ in the sleep phase doesn't necessarily encourage an improvement to the ELBO, which directly relates to the compression rate in the lossless compression application [38]. Therefore, in the next section, we propose a new variational inference scheme: *reverse sleep amortized inference* and demonstrate how it helps improve the generalization of the inference network in practice.

## 3.2 Reverse Sleep Amortized Inference

We propose to use the *reverse KL divergence* in the sleep phase. We fix $\theta$ and train $\phi$ using

$$\min_\phi \left\langle \mathrm{KL}(q_\phi(z|x)||p_\theta(z|x) \right\rangle_{p_\theta(x)} = \max_\phi \left\langle \log p_\theta(x, z) - \log q_\phi(z|x) \right\rangle_{q_\phi(z|x)p_\theta(x)}, \tag{19}$$

where the integration $\langle \cdot \rangle_{p_\theta(x)}$ is approximated by Monte-Carlo using samples from the generative model $p_\theta(x)$. This reverse KL objective encourages improvements to the ELBO. When we have a perfect model $p_{\theta*}(x) = p_d(x)$ the reverse sleep phase is equivalent to

$$\min_\phi \left\langle \mathrm{KL}(p_{\theta*}(z|x)||q_\phi(z|x)) \right\rangle_{p_{\theta*}(x)} = \min_\phi \left\langle \mathrm{KL}(p_{\theta*}(z|x)||q_\phi(z|x)) \right\rangle_{p_d(x)} \tag{20}$$

which satisfies the *inference consistency requirement*.

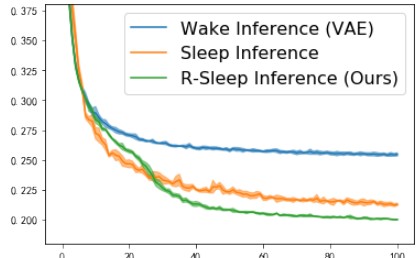

Figure 3: Test BPD vs epochs. We compare the consistency property between three amortized inference methods.

The consistency requirement can also be validated empirically when the perfect model is known $p_{\theta*}(x) = p_d(x)$. This can be achieved by using a pre-trained VAE as the true data generation distribution. Therefore, we first train a VAE to fit the binary MNIST problem. The VAE has the same structure as that used in Section 2 and is trained using Adam with $lr = 1 \times 10^{-3}$ for 100 epochs. After training, we treat the pre-trained decoder $p_{\theta'}(x|z)$ as the training data generator $p_d(x) \equiv \int p_{\theta'}(x|z)p(z)dz$. We then sample 10000 data samples from $p_d$ to form a training set $\mathcal{X}_{train}$ and 1000 samples to form a test set $\mathcal{X}_{test}$. We then train a new $q_\phi(z|x)$ with: 1) wake phase inference (VAE) 2) (forward) sleep inference and 3) reverse sleep inference. The network is trained using Adam with $lr = 1 \times 10^{-3}$ for 100 epochs. Figure 3 shows the test BPD calculated after every training epoch. We can see the sleep phase out-performs the wake phase and the reverse sleep inference achieves the best BPD. Intuitively, this is because both the forward and reverse sleep inference use the true model to generate additional training data whereas the wake inference only has access to the finite training dataset $\mathrm{X}_{train}$.

### 3.3 Reverse Half-asleep Amortized Inference with Imperfect Models

In practice our model will not be perfect $p_\theta \neq p_d$. Empirically we find that samples from even a well trained model $p_\theta$ may not always be sufficiently like the samples from the true data distribution. This can lead to degradation in the performance of the inference network when using the reverse-sleep approach. For this reason, we propose to use a mixture distribution between the model and the empirical training data distribution as follows

$$\big\langle \mathrm{KL}\big(q_\phi(z|x)||p_\theta(z|x)\big)\big\rangle_{m(x)} \quad \text{where} \quad m(x) \equiv \alpha p_\theta(x) + (1-\alpha)\hat{p}_d. \tag{21}$$

When $\alpha = 0$, it reduces to the standard approach used in VAE training. When $\alpha = 1$, we recover the reverse sleep method (Equation 19). We find that a setting of $\alpha = 0.5$ works well in practice. This balances samples from the true underlying data distribution with samples from the model.

We thus refer to this method as *reverse half-asleep* since it uses both data and model samples to train the amortized posterior. Intuitively, we can rewrite the Equation 21 as a sum of two positive terms

$$\alpha\big\langle \mathrm{KL}\big(q_\phi(z|x)||p_\theta(z|x)\big)\big\rangle_{\hat{p}(x)} + (1-\alpha)\big\langle \mathrm{KL}\big(q_\phi(z|x)||p_\theta(z|x)\big)\big\rangle_{p_\theta(x)}. \tag{22}$$

Therefore, the optimal of this objective will make the first term 0, which is the same requirement as the classic amortized inference (Equation 8). The second term, which is equivalent to the reverse sleep amortized inference (Equation 19), can encourage the inference consistency requirement: when $p_\theta = p_d$, the optimal of the second term will set $q_\phi(z|x) = p_\theta(z|x)$ for any $x \sim p_d(x)$. When $p_\theta$ is not perfect, the second term can be seen as a regularizer added to the classic amortized inference objective, which can be used to penalize the hypothesis space of the amortized network [32].

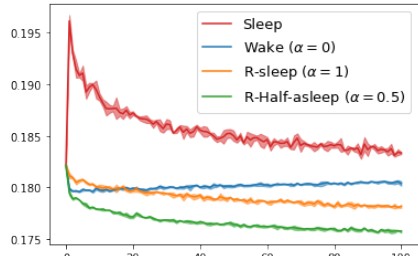

To compare with different $\alpha$, we first fit a VAE (with the same structure as that used in Figure 2) to the Binary MNIST dataset, and then train the amortized posterior using sleep inference (Equation 17) and three different $\alpha$ for additional 100 epochs using Adam with learning rate $3 \times 10^{-4}$. Figure 4 shows the test BPD comparison. We find the proposed reverse half-asleep method ($\alpha = 0.5$) outperforms the reversed sleep method ($\alpha = 1$), whereas the standard amortized inference training in VAE ($\alpha = 0$) leads to overfitting of the inference network. We also plot the sleep inference training curve, whose BPD is less competitive since it is not directly optimizing the ELBO.

Figure 4: Test BPD comparisons of Amortized inference with different $\alpha$. We find the Reverse Half-asleep method ($\alpha = 0.5$) achieves the best BPD. The mean and std are calculated with three random seeds.

## 4 Generalization Experiments

We apply the reverse half-asleep to improve the generalization of VAEs on three different datasets: binary MNIST, grey MNIST [24] and CIFAR10 [23]. For binary and grey MNIST, we use latent dimension 16/32 and neural nets with 2 layers of 500 hidden units in both the encoder and decoder. We use Bernoulli $p(x|z)$ for binary MNIST and discretized logistic distribution for grey MNIST. We train the VAE with the usual amortized inference approach using Adam with $lr = 3 \times 10^{-4}$ for 1000 epochs and save the model every 100 epochs. We then use the saved models to 1) evaluate on the test data sets, 2) conduct optimal inference by training $q_\phi(z|x)$ on the test data and 3) run reverse half-asleep method before calculating the test BPD. For the reverse half-asleep, we train the amortized posterior for 100 epochs with Adam and $lr = 5 \times 10^{-4}$. To sample from $p_\theta(x)$, we firstly sample $z' \sim p(z)$ and sample $x' \sim p(x|z = z')$. For the optimal inference strategy, we train the amortized posterior with the same optimization scheme on the test data set for additional 500 epochs to ensure the same number of gradient steps are conducted (since training set is 5 times as big as the test set). Figure 5a and 5b show the test BPD comparisons of binary and grey MNIST respectively and demostrate that our approach does not require further training on the test data to improve generalization performance.

For CIFAR10, we use the convolutional ResNet [17, 40] with 2 residual blocks and latent size 128. The observational distribution is a discretized logistic distribution with linear autoregressive

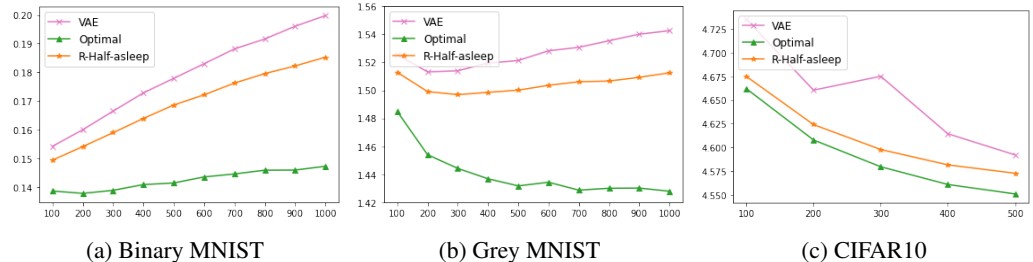

(a) Binary MNIST          (b) Grey MNIST          (c) CIFAR10

Figure 5: Test BPD comparisons among amortized inference (VAE), optimal inference strategy and the reverse half-asleep inference on three datasets. The x-axis represents the training epochs.

parameterization within channels. We train the VAE for 500 epochs with Adam and $lr = 5 \times 10^{-4}$ and save the model every 100 epoch. The pre-trained VAE achieves 4.592 BPD on the CIFAR10, which is comparable with other single latent VAE models reported in [40]: 4.51 BPD with a VAE with latent dimension 256 and 4.67 BPD with a discrete latent VAE (VQVAE).

Ideally, when the VAE model converges to the true distribution $p_\theta \to p_d$, the aggregate posterior $q_\phi(z) = \int q_\phi(z|x)p_d(x)dx$ will match the prior $p(z)$. However, for a complex distribution like CIFAR10, a significant mismatch between $q_\phi(z)$ and $p(z)$ is usually observed in practice [51, 12]. In this case, the sample $x'$ that is generated using a latent sample from the prior $x' \sim p_\theta(x|z')$, where $z' \sim p(z)$, may be blurry or invalid. A common solution is to train another model, e.g. a VAE [12] or a PixelCNN [41, 40] to approximate $q_\phi(z)$. In our case, we instead directly sample from $q_\phi(z)$ rather than $p(z)$ to generate samples in Equation 19, which can be done by first sampling $x' \sim p_d(x)$ (from the training dataset) and then sample $z' \sim q_\phi(z|x = x')$. This scheme still results in a consistent training objective since $q_{\phi^*}(z) = p(z)$ for the optimal posterior $q_{\phi^*}(z|x)$. We use Adam with $lr = 1 \times 10^{-5}$ and train the reverse half-asleep inference for 100 epochs on the training data and train the optimal inference strategy for 500 epochs on the test data, see Figure 5c for the result. We find the proposed reverse half-asleep training approach (with sampling from $q_\phi(z)$) consistently improves the generalization performance of the amortized posterior. We also apply the proposed method on a VAE trained on CIFAR100 for 500 epochs (the rest of the experiment settings are the same as the CIFAR10 case) and find our method improves the BPD from 5.288 to 5.275.

## 4.1 Comparisons with Regularization Methods

Recent work [35] proposed to alleviate overfitting of amortized inference by optimizing a linear combination between the traditional amortized inference (Equation 8) and a denoising objective

$$\alpha \langle \mathrm{KL}(q_\phi(z|x + \epsilon) || p_\theta(z|x)) \rangle_{p(\epsilon)} + (1 - \alpha)\mathrm{KL}(q_\phi(z|x) || p_\theta(z|x)), \tag{23}$$

where $p(\epsilon) = \mathcal{N}(0, \sigma^2 I)$. We compare this regularizer to our method by training the amortized posterior of VAEs for an additional 100, 300 and 100 epochs on Binary, Grey MNSIT and CIFAR respectively. For the denoising regularizer, we use the same linear combination weight $\alpha = 0.5$ as that used in Equation 21 and vary $\sigma \in \{0.1, 0.2, 0.4, 0.6, 0.8, 1.0\}$, see Table 1 for the comparisons. For MNIST, we find $\sigma \in \{0.1, 0.2, 0.4\}$ improves the generalization but larger noise levels hurts the performance. For CIFAR10, only $\sigma = 0.1$ can slightly improve the generalization by 0.001 BPD. In contrast, our method consistently achieves better generalization performance without tuning any hyper-parameters, see Figure 6 for the test BPD (evaluated every training epoch, the mean/std are calculated with 3 random seeds). Compared to the denoising approach, one limitation of our method is the requirement of model samples, which is more computational expansive during training.

Since the decoder is shared and fixed in all comparisons, better test ELBO indicates the predicted $q_\phi(z|x')$ is closer to the true posterior $p_\theta(z|x')$ under the KL divergence measure (see Equation 4, higher ELBO with fixed $\theta$ indicates $\mathrm{KL}(q_\phi(z|x) || p_\theta(z|x))$ is smaller). Therefore, the proposed method can also benefit a range of tasks that require accurate prediction of the posterior on the test data. In Appendix A and B, we demonstrate our method can provide better proposal distributions for the importance weighted Auto-Encoder [7] and also improve the representation learning performance for down-stream classification tasks.

Table 1: Average test BPD comparisons with Denoising Regularizer [35].

| Methods | VAE | $\sigma = 0.1$ | $\sigma = 0.2$ | $\sigma = 0.4$ | $\sigma = 0.8$ | $\sigma = 1.0$ | Ours |
|---|---|---|---|---|---|---|---|
| Binary MNIST | 0.200 | 0.195 | 0.192 | 0.191 | 0.196 | 0.201 | **0.187** |
| Grey MNIST | 1.543 | 1.527 | 1.519 | 1.515 | 1.545 | 1.550 | **1.513** |
| CIFAR10 | 4.592 | 4.591 | 4.598 | 4.614 | 4.651 | 4.667 | **4.572** |

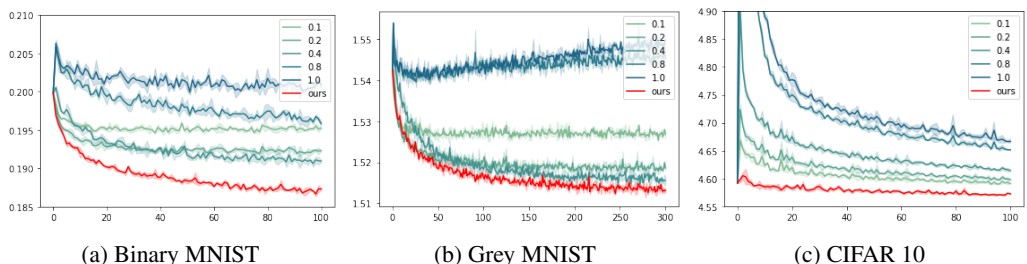

(a) Binary MNIST         (b) Grey MNIST         (c) CIFAR 10

Figure 6: Test BPD evaluated after every training epoch. We find, compared to the denoising regularizer, the proposed amortized inference training scheme consistently achieves better generalization performance in all tasks.

## 5 Application of Lossless Compression

Lossless compression is an important application of VAEs where generalization plays a key role in the compression rate. Given a VAE with $p_\theta(x|z)$, $q_\phi(z|x)$ and $p(z)$, a practical compressor can be efficiently implemented using the Bits Back algorithm [19, 38] with the ANS coder [14]. See Appendix E for a detailed introduction of conducting lossless compression with VAE models. In Algorithm 1, we summarize the Bits Back procedure with amortized inference to compress/decompress a test data point $x'$ to a stack that contains bit string messages. The resulting code length for data $x'$ is approximately equal to the negative ELBO

$$- \log_2 p_\theta(x'|z') - \log_2 p(z') + \log_2 q_\phi(z'|x'). \tag{24}$$

We have shown that $q_\phi(z|x)$ may overfit to the training data, degrading compression performance. To improve the compression BPD, the optimal inference strategy can also be applied in the Bits Back algorithm. In the compression stage, we can train $\phi$ by

$$\phi^* = \arg\max_\phi \text{ELBO}(x', \theta, \phi). \tag{25}$$

When the $q_\phi(z|x')$ is parameterized to be a Gaussian, we can just take $\phi$ to be the mean and standard deviation $\mathcal{N}(\phi_\mu, \phi_\sigma^2)$, which only contains two training parameters. In

---

**Algorithm 1** Bits Back with Amortized Inference.

Comp./decomp. stages share $\{p_\theta(x|z), q_\phi(z|x), p(z)\}$.

———————— Compression ————————
Draw sample $z' \sim q_\phi(z|x')$ from the stack.
Encode $x' \sim p_\theta(x|z')$ onto the stack.
Encode $z' \sim p(z)$ onto the stack.

———————— Decompression ————————
Decode $z' \sim p(z)$ from the stack.
Decode $x' \sim p_\theta(x|z')$ from the stack.
Encode $z' \sim q_\phi(z|x')$ onto the stack.

---

the decompression stage, we observe that the compressed data $x'$ is recovered before the $q_\phi(z|x')$ is used to encode $z'$. Therefore, we can also train the $q_\phi(z|x')$ using the recovered $x'$ to maximize the test ELBO. If the optimization procedure is the same as that used in the compression stage, we will get the same $q_{\phi^*}(z|x')$. In practice, we need to pre-specify the number of gradient descent steps $K$. When $K$ is large, we recover the optimal inference strategy and the code length is approximately

$$- \log_2 p_\theta(x'|z') - \log_2 p(z') + \log_2 q_{\phi^*}(z'|x'). \tag{26}$$

This observation was first proposed in [45] in the context of lossy compression and then applied to lossless compression with Bits Back coding in [30]. Furthermore, by varying the optimization steps $K$ in the optimal inference, we can trade off between the speed and the compression rate. This is valuable for practical applications with different speed/rate requirements. See Algorithm 2 for a summary of the Bits Back algorithm with $K$-step optimal inference.

Although the optimal inference strategy can be used in lossless compression, it requires extra run-time for training at the compression stages. In contrast, our proposed reverse half-asleep inference scheme can improve the compression rate *without* scarifying any speed. Additionally, our method can also provide a better initialization for the optimal-inference strategy to allow a better trade-off between compression rate and speed.

We implement[4] Bits Back with ANS [14] and compare the compression among four inference methods:

---

**Algorithm 2** Bits Back with $K$-step Optimal Inference

Comp./decomp. stages share $\{p_\theta(x|z), q_\phi(z|x), p(z)\}$ and the optimization procedure of Equation 25.

——————————— Compression ———————————

Take K gradient steps $\phi \to \phi^K$ with Equation 25.
Draw sample $z' \sim q_{\phi^K}(z|x')$ from the stack.
Encode $x' \sim p_\theta(x|z')$ onto the stack.
Encode $z' \sim p(z)$ onto the stack.

——————————— Decompression ———————————

Decode $z' \sim p(z)$ from the stack.
Decode $x' \sim p_\theta(x|z')$ from the stack.
Take K gradient steps $\phi \to \phi^K$ with Equation 25.
Encode $z' \sim q_{\phi^K}(z|x')$ onto the stack.

---

**1. Baseline:** This is the classic VAE-based compression introduced by [38]. For binary and grey MNIST, both the encoder and decoder contain 2 fully connected layers with 500 hidden units and latent dimension 10. The observation distributions are Bernoulli and discretized Logistic distribution respectively. For CIFAR10, we use fully convolutional ResNets [17] with 3 residual blocks in the encoder/decoder, latent dimension 128 and discreteized Logistic distribution with channel-wise linear autoregressive[31] as the observation distribution. We train both the amortized posterior and the decoder by maximizing the ELBO (Equation 3) using Adam with $lr = 3 \times 10^{-4}$ for 100, 100 and 500 epochs (for Binary MNIST, Grey MNIST and CIFAR10 respectively), and then apply Algorithm 1 to conduct compression.

**2. Reversed Half-asleep**: we do amortized inference using Equation 21 for 100 and 300 epochs with Adam optimizer ($lr = 3 \times 10^{-4}$) for binary and grey MNIST respectively, and $lr = 1 \times 10^{-5}$ for 100 epochs for CIFAR10. Other training details are the same as the baseline method.

**3. Optimal Inference:** we take the amortized posterior (encoder) and decoder from the baseline and apply the $K$-step optimal inference strategy described in Algorithm 2 to do compression. We use Adam optimizer and vary the $K$ from 1 to 10 to achieve a trade-off curve between compression rate and speed. We actively choose the highest learning rate that can make the BPD consistently improve with the increment of $K$: $lr = 5 \times 10^{-3}$ for binary and grey MNIST and $lr = 1 \times 10^{-3}$ for CIFAR10.

**4. Reversed Half-asleep + Optimal Inference:** we take the encoder in method 2 and decoder from the baseline and conduct $K$-step optimal inference. All other training details are as per method 3.

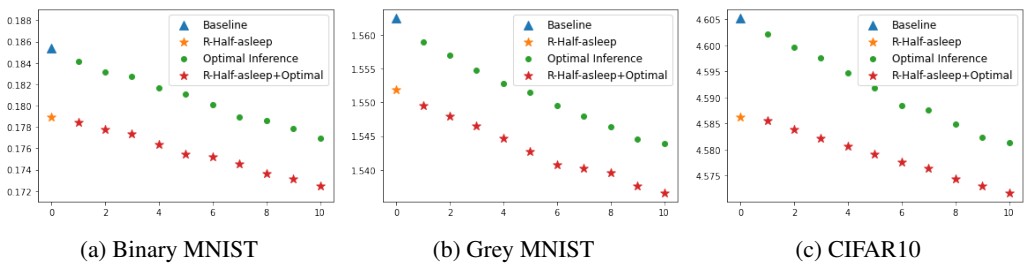

(a) Binary MNIST        (b) Grey MNIST        (c) CIFAR10

Figure 7: We plot the comparisons for different methods. The y-axis is the BPD and x-axis represents the $K$ gradient steps in the optimal inference. The baseline and our R-Half-sleep can be seen as special cases of optimal inference with $K = 0$. We find given a fixed computational budget, our method achieves a lower BPD than one using traditional amortized inference training.

In Figure 7, we plot test BPD comparisons for the different methods outlined. We can see if optimization is not allowed at compression time, the use of our reverse-half-asleep method achieves better compression rate with no additional computational cost. If we allow $K$-step optimization during compression, for a given computational budget, the amortized posterior initialized using our

---

[4]Implementation can be found in the following repo: `https://github.com/zmtomorrow/GeneralizationGapInAmortizedInference`. All experiments are run on a NVIDIA V100 GPU.

| | Baseline | Ours | K=7 |
|---|---|---|---|
| BPD | 0.185 | 0.179 | 0.179 |
| Com. Time | 0.006 | 0.006 | 0.013 |
| Dec. Time | 0.006 | 0.006 | 0.013 |
| Time Cost | - | 0% | 116.7% |

(a) MNIST

| | Baseline | Ours | K=8 |
|---|---|---|---|
| BPD | 4.602 | 4.585 | 4.585 |
| Com. Time | 0.27 | 0.27 | 0.38 |
| Dec. Time | 0.26 | 0.26 | 0.38 |
| Time Cost | - | 0% | 46.2% |

(b) CIFAR10

Figure 8: Compression (Com.) and decompression (Dec.) time comparison. We show that to achieve the same BPD as our method, the $K$-step optimal inference strategy that initializes the amortized posterior needs $K = 7$ (binary MNIST) and $K = 8$ (CIFAR10) steps for each test datapoint, which will cost an additional $116.7\%$ and $46.2\%$ of time respectively during compression.

reverse-half-asleep method also achieves lower BPD, which leads to a better trade-off between the time and compression rate. Table 8 also reports the average time improvements of our method to compress a single MNIST and CIFAR10 image respectively, which shows the effectiveness of our method.

## 6  Related Work

A different perspective on generative models' generalization is proposed in paper [50] where the generalization is evaluated by testing if the model can generate novel combinations of features. However, the generalization defined in our work is purely measured by the test likelihood, which is a different perspective and more relevant for the application of lossless compression.

Recent work [49] first studies the likelihood-based generalization for lossless compression. They focus on the test and train data that are from different distributions whereas we assume they follow the same distribution. Additionally, their model has a tractable likelihood and relates to the *generative model related generalization*, whereas we focus on *inference related generalization* in VAEs.

Previous work [11] studied the *amortization gap* in amortized inference, which is caused by using $q_{\phi^*}(z|x_n)$ to generate posteriors for each input $x_n$ rather than learning a posterior $q_n^*(z)$ for $x_n$ individually. This gap can be alleviated using a larger capacity encoder network. This amortization gap is fundamentally different from the *inference generalization gap* we discuss in this work since the latter focuses solely on test time generalization but the former problem also exists at training time.

Recent work [30] proposes a compression scheme based on the IWAE [7] bound, which is tighter than the ELBO and thus improves the compression rate. However, this method has to compress/decompress multiple latent samples, which requires extra time cost. On the other hand, we focus on improving the ELBO-based compression that only needs to compress one single latent sample. Nevertheless, similar to the $K$-step optimal inference strategy, our amortized training objective can also be used in the IWAE-based method, which gives a better proposal distribution for importance sampling, see Appendix A for a demonstration.

Paper [8] considers the following data generation procedure $x_1 \sim p_d(x)$, $z_1 \sim p_\theta(z|x_1)$, $x_2 \sim p_\theta(x|z_1)$ and propose to enforce latent consistency between $q_\phi(z|x_1)$ and $q_\phi(z|x_2)$ for paired data $(x_1, x_2)$ to encourage the robustness of the learned representation. This procedure is close to the self-supervised contrasting learning method [10] where the augmented data is the reconstruction of the training data using the VAE model. In our paper, we want to encourage the sample from the model $x' \sim \int p_\theta(x|z)p(z)dz$ to have high ELBO under the model (Equation 19) to improve the generalization of the amortized inference and no paired data is required in our procedure. Therefore, both motivations and methodologies are different from our method.

## 7  Conclusion

We have shown how the generalization of VAEs is largely affected by the amortized inference network and proposed a new variational inference scheme that provides better generalization as demonstrated in the application of lossless compression. Future work will study the generalization of the decoder model to further improve the performance of VAEs.

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
