# A Tightness of the ELBO and IWAE Improvement

In this section we want to verify the tightness of the ELBO as a lower bound of the log likelihood. Consider the likelihood for a single data point $x'$, we have

$$\log p_\theta(x') \geq \langle \log p_\theta(x'|z) \rangle_{q_\phi(z|x')} - \mathrm{KL}(q_\phi(z|x')||p(z)) \equiv \mathrm{ELBO}(x, \theta, \phi). \qquad (27)$$

To evaluate $\log p_\theta(x')$, we can use an importance weighted estimation (IWAE [7]), which can be rewritten as

$$\log p_\theta(x') = \log \left\langle \frac{p_\theta(x'|z)p(z)}{q_\phi(z|x)} \right\rangle_{q_\phi(z|x)} \approx \log \frac{1}{K} \sum_{k=1}^{K} \frac{p_\theta(x'|z_k)p(z_k)}{q_\phi(z_k|x')} \equiv \mathrm{IWAE}_k(x, \theta, \phi), \quad (28)$$

where $z_k \sim q_\phi(z|x')$. The accuracy of the importance sampling heavily depends on the proposal distribution $q_\phi(z|x')$ and will be poor if $q_\phi(z|x')$ underestimates the high density region of $p_\theta(z|x)$ [7]. For the ELBO with optimal inference, we can assume the approximate posterior is close to the true posterior, so if the lower bound is tight, we will observe that the ELBO is approximately equal to the IWAE. In Figure 9 we compare the ELBO and IWAE using classic amortized inference and optimal inference respectively (we use $k = 10$ in all cases). We find that the IWAE can improve the ELBO for the traditional amortized inference and is approximately equivalent to the ELBO using the optimal inference strategy. Therefore, we can conclude that the ELBO with the optimal inference strategy is tight to $\log p_\theta(x)$.

We also estimate the IWAE using the proposal posterior learned by the proposed reverse half-asleep inference and find that our method can also improve the IWAE result, see Figure 9 for details. This is intuitive since our method can provide a better proposal distribution for importance sampling.

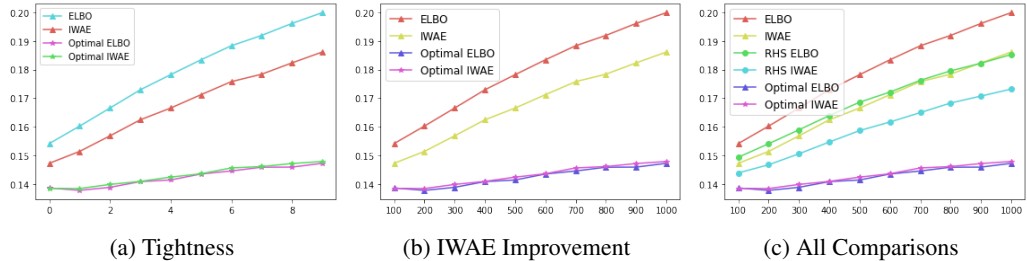

(a) Tightness       (b) IWAE Improvement       (c) All Comparisons

Figure 9: IWAE comparisons on Binary MNIST. The x-axis indicates the training epoch and the y-axis is the Bits-per-dimension, which corresponds to the negative ELBO or IWAE with log 2 base and normalized by data dimension, lower is better. In Figure a, we see that IWAE improves the ELBO when using classic amortized inference but is approximately equal to the ELBO when using optimal inference, which indicates the bound is tight. In Figure b, we compare the IWAE with classic amortized inference, optimal inference and the the proposed reverse half-asleep (RHS) inference. Here we find the proposed method can also improve the classic IWAE estimation without training on the test data. In Figure 3, we plot the ELBO and IWAE for all three amortized inference methods.

# B Amortized Posterior for Down-stream Classification Task

In Section 4, we discussed that the proposed reverse half-asleep method can improve the posterior prediction for the test data. One direct application is to use the learned amortized posterior $q_\phi(z|x)$ for down-stream tasks, e.g. image classification, where the samples $z' \sim q_\phi(z|x')$ can be treated as the 'stochastic representation' [48, 4] of the given data point $x'$. Given a labeled dataset $\{(x_1, y_1), \cdots, (x_N, y_N)\}$ and a trained amortized posterior (encoder) $q_\phi(z|x)$, we can then train a classifier $p_\eta(y|z)$ that maps from the latent space $z$ to the label $y$. After training the classifier, for a given test set of unlabelled data $\{x'_1, \cdots, x'_M\}$, the predictive distribution can be written as $p(y|x) = \int p_\eta(y|z)q_\phi(z|x)dz$ and can be approximated by Monte-Carlo: $p(y|x) \approx \frac{1}{K} \sum_{k=1}^{K} p(y|z'_k)$, where $z'_k \sim q_\phi(z|x)$. We train a classifier with 2 layer feed-forward neural network with hidden size 200, RelU activation and dropout with rate 0.1 on two datasets: binary MNIST and grey MNIST. The models are trained for 10 epochs with Adam optimizer and learning

rate $3\times10^{-4}$. During training, we randomly sample one $z'$ for each data point $x$ and we use $k=100$ in the testing stage to estimate the predictive distribution. Figure 10 shows the comparisons between the posterior trained by the classic amortized inference and the proposed reverse half-asleep method respectively. We can see our method consistently improves the classification accuracy performance.

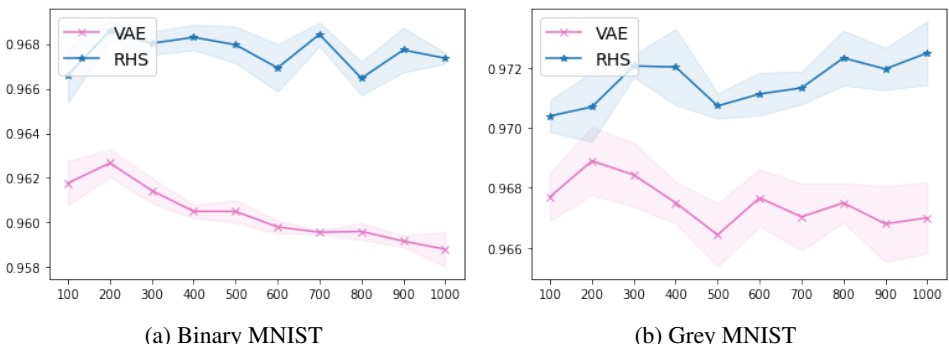

(a) Binary MNIST       (b) Grey MNIST

Figure 10: Representation Learning for Down-Stream Classification. We train the VAE for 1000 epochs and evaluate the classification accuracy (y-axis, higher is better) on the down-stream classification task every 100 epochs (x-axis). The results are averaged over 3 random seeds and we also plot the standard deviation.

## C    Effects of the Latent Space Dimensionality

We study the effect of the latent dimension size on the generalization of the amortized inference. We use the VAE described in Section 4 with different latent size $[16, 64, 128]$ on Binary MNIST, see Figure 11 for the result. We find the overfitting of amortized inference happens in all cases regardless of the latent size. We also apply the proposed reverse half-asleep training method to the saved model every 100 epoch and found our method can consistently improve the generalization performance.

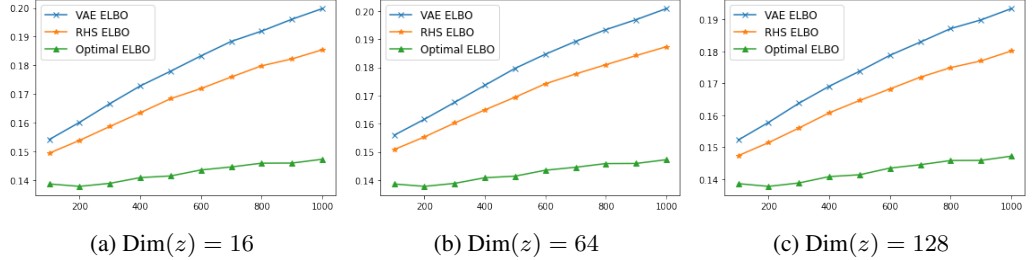

(a) $\mathrm{Dim}(z) = 16$       (b) $\mathrm{Dim}(z) = 64$       (c) $\mathrm{Dim}(z) = 128$

Figure 11: Effects of different latent dimension. The y-axis is the BPD and x-axis is the training epochs. We find the amortized inference generalization gap exits in all cases.

## D    Reverse Half-asleep From the Beginning

In section 4 we applied the reverse half-asleep training in a post-hoc fashion, which allow us to isolate the degree to which both the generative model and amortized inference generalization gap are contributing to overfitting. It has also been observed that a poor variational posterior in the early stage of the training will cause the M-step of the generative model $p_\theta(x|z)$ to get trapped into a local minimum ( see "Two problems with variational expectation maximization for time-series models" section in [3]). Therefore, we can also apply the proposed method from the beginning of training, see Figure 12 for the results. For a simple dataset like Binary MNIST, we find that using the proposed reverse half-asleep from the beginning can lead to a better test ELBO compared to the classic VAE training, or our proposed post-hoc training. However, for a more complex dataset like grey-scale MNIST, the result of using the reverse half-asleep from the beginning is worse than the classic VAE training. We hypothesize that for a complex dataset, the decoder in the beginning cannot generate

valid images, which will lead to biased gradients. Therefore, we also report the result of using the reverse half-asleep training starting from 200 epochs onwards and find it is better than the classic VAE training but is worse than the post-hoc reverse half-asleep method. We leave the study of how to improve the generalization from the beginning of the training to future work.

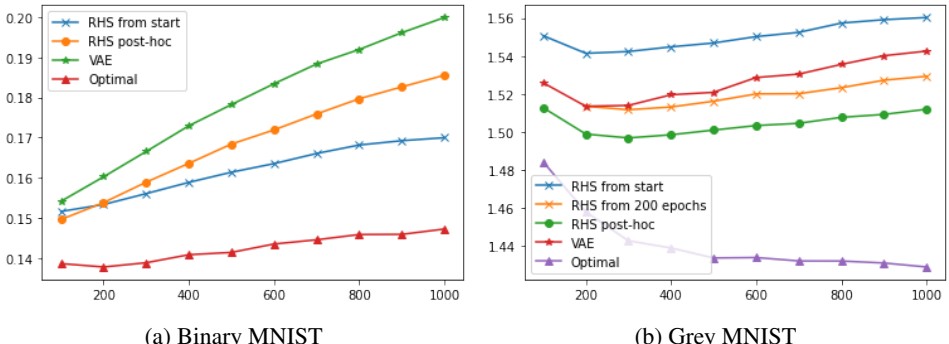

(a) Binary MNIST          (b) Grey MNIST

Figure 12: We compare different ways of using the proposed training objective (from the beginning or post hoc). We also plot the standard VAE training and the ELBO with optimal inference for reference.

# E    Introduction of Bits Back Coding with VAE

In this section, we briefly introduce the background of the model-based lossless compression and the Bits Back coding scheme of VAE models.

## E.1    Model-based Lossless Compression

The goal of the lossless compression is to create an invertible mapping from real-world data (e.g. image, audio, video) to binary strings with the lengths of the strings as short as possible.

Let $X$ be a discrete random variable that taking values from a finite countable set $\mathcal{X}$ and has a probability mass function (PMF) $p : \mathcal{X} \to \mathbb{R}$ such that $\forall x \in \mathcal{X}, p(x) > 0$ and $\sum_x p(x) = 1$.

**Definition 1** *The Shannon information content of a sample $x \sim p(x)$ is defined as*

$$h_p(x) \equiv - \log_2 p(x). \tag{29}$$

**Definition 2** *The Shannon Entropy of a distribution $p$ is defined as*

$$H(p) \equiv - \sum_x p(x) \log_2 p(x). \tag{30}$$

We then give the informal statement of the *Shannon Source Coding Theorem* [33], the detailed statement and the proof can be found in Chapter 4 of [27].

**Theorem 1 (Shannon's Source Coding Theorem (informal))** *$N$ i.i.d samples form the data generation distribution with PMF $p_d(x)$ can be losslessly compressed into more than $NH(p_d)$ bits when $N \to \infty$. Conversely, they cannot be losslessly compressed into fewer than $NH(p_d)$ bits.*

To obtain a 'near-optimal' lossless compression scheme in practice, one strategy is to compress each data $x \sim p_d(x)$ into a binary string with length equal to $h_{p_d}(x) + \epsilon$, where $h(x)$ is the *Shannon information content* and $\epsilon$ represents a small coding overhead. Therefore, given $N$ i.i.d samples $\{x_1, \cdots, x_N\} \sim p_d(x)$, the averaged compression length is

$$-\frac{1}{N} \sum_{n=1}^{N} \log_2 p_d(x_n) + \epsilon \xrightarrow{N \to +\infty} - \sum_x p_d(x) \log_2 p_d(x) + \epsilon = H(p_d) + \epsilon, \tag{31}$$

which is close to optimal when $\epsilon$ is small.

Different coders are proposed to make the overhead $\epsilon$ for different types of data. For multi-dimensional data, there exits two methods that can provide us 'near-optimal' lossless compression: Arithmetic Coding (AC) [44] and Asymmetric Numeral System (ANS) [14], we recommend Chapter 6 of [27] and [37] for detailed introductions of the two methods respectively. We use the ANS coder in this paper since it has a faster speed comparing to AC. For simplicity, we abstract an ANS coder as an invertible function $\text{enc}_p(\cdot)$ that maps a given data $x' \in \mathcal{X}$ to a binary string message $m'$ with length $\text{len}(m') = -\log_2 p(x') + \epsilon$, where $\epsilon$ is a negligible coding overhead. We also denote the decoding function as $\text{dec}_p(\cdot) = \text{enc}_p^{-1}(\cdot)$ and have $\text{dec}_p(m') \to x'$.

We have introduced how to optimally compress the data when we know the true data generation distribution $p_d(x)$. However, the distribution $p_d(x)$ is usually unknown in practice, we would like to learn a model $p_\theta(x)$ to approximate the underlying data distribution $p(x)$ and then use the learned model $p_\theta(x)$ to conduct lossless compression. In this case, the *averaged* data compression length for $\{x_1, \cdots, x_N\} \sim p_d(x)$ is (ignoring the coding overhead $\epsilon$):

$$-\frac{1}{N} \sum_{n=1}^{N} \log_2 p_\theta(x_n) \xrightarrow{N \to +\infty} -\sum_x p(x) \log_2 p_\theta(x_n). \tag{32}$$

The difference between the model compression length and the optimal compression length is

$$-\frac{1}{N} \sum_{n=1}^{N} \Big( \log_2 p_\theta(x_n) - \log_2 p_d(x_n) \Big) \xrightarrow{N \to +\infty} \text{KL}(p_d(x)||p_\theta(x)). \tag{33}$$

### E.2   Bits Back Compression with VAEs

Given a discrete Latent VAE model specified by the PMFs $\{p_\theta(x|z), q_\phi(z|x), p(z)\}$ and a target data $x'$ to compress. A naive strategy is to first generate a sample $z' \sim q_\phi(z|x')$ and then encode $x'$ with $p_\theta(x|z')$. We also encode $z'$ with distribution $\log p(z)$, so the total code length is then We also encode $z'$ with distribution $\log p(z)$, so the total code length is then

$$-\log_2 p_\theta(x'|z') - \log_2 p(z'), \tag{34}$$

which is larger than the optimal code length $-\log_2 p(x')$ by $-\log_2 p_\theta(z'|x')$ bits. To achieve the optimal code length, a key observation is that the sampling process $z' \sim q_\phi(z|x')$ can be done by decoding random bits using the distribution $q_\phi(z|x')$. Specifically, we assume that we can access a message that already contains random bits, which we visualize as the following figure[5].

Initial random bits

**In the encoding stage**, we first sample $z'$ form $q_\phi(z|x')$ by decoding random bits with distribution $q_\phi(z'|x')$, so the message length decreased by length $-\log_2 q_\phi(z'|x')$.

We then encode $x'$ with distribution $p_\theta(x'|z')$, so the message is increased by length $-\log_2 p_\theta(x'|z')$.

Finally, we encode $z'$ with distribution $p(z)$ and the message length is increased by $-\log p(z')$.

**In the decoding stage**, we first decode $z'$ using $p(z)$.

---

[5]The visualization is taken from [38].

$$-\log_2 p(z')$$

$$\xrightarrow{\mathrm{dec}_{p(z)}(\cdot)} \quad z'$$

We then decode $x'$ with distribution $p_\theta(x|z')$.

$$-\log_2 p_\theta(x'|z')$$

$$\xrightarrow{\mathrm{dec}_{p_\theta(x|z')}(\cdot)} \quad x'$$

Finally, we encode the random bits 'back' to the stack to recover the initial message.

$$-\log_2 q_\phi(z'|x')$$

$$\xleftarrow{\mathrm{enc}_{q_\phi(z|x')}(\cdot)} \quad z'$$

Therefore, the 'net' message length to compress data $x'$ with a VAE is equivalent to

$$-\log_2 p_\theta(x|z') - \log_2 p(z') + \log q_\phi(z'|x'), \tag{35}$$

which is a one-sample estimation of the ELBO and is optimal (equal to $-\log_2 p_\theta(x')$) when the amortized variational posterior is equal to the true posterior $q_\phi(z|x') = p_\theta(z|x')$.

This scheme and also be extended to continuous latent $z$ with negligible cost by quantizing the PDF $p(z)$ and $q_\phi(z)$ into PMF to conduct the compression. See [38] for details. This 'Bits Back' coding method was first introduced as a thought experiment in [42, 19] and was later implemented by [15] with an AC coder. Recently, [38] proposed to implement the Bits Back with ANS [39] and a VAE model, which allows great improvement of both the compression rate and the computational efficiency. We refer the reader to [38] for other practical considerations and implementation details.