# OpenReview forum: "Generalization Gap in Amortized Inference"
_NeurIPS.cc/2022/Conference — NeurIPS 2022 Accept_

### Official Review · Reviewer_rcEf · 2022-07-07

**Rating:** 6
**Confidence:** 3
**Ethics Flag:** Yes
**Soundness:** 3 good
**Presentation:** 3 good
**Contribution:** 3 good

**Summary:**

This paper focuses on the generalization gap in amortized inference trained using the ELBO objective. The generalization gap can come from either the generative model p or the inference network q not generalizing from training data to test data. This paper argues that q is mainly responsible for the generalization gap, which is supported by an experiment using the VAE model and the MNIST dataset (Figures 1 and 2). The proposed fix is to train on $E_{m(x)}[ELBO]$ instead of $E_{\hat p(x)}[ELBO]$ where $m(x) = \alpha p_\theta(x) + (1 - \alpha) \hat p(x)$, i.e. "augment" the empirical dataset by samples from a well-trained model. Experiments suggest that this indeed leads to better test ELBOs, which in turn leads to better compression rates.

I gave this paper a 5 but am willing to increase the score if the authors address some of the comments below.

**Questions:**

I'd also appreciate authors' thoughts on the following points.

The paper claims that KL(q || p) is better than KL(p || q) because of compression. However, KL(p || q) also comes with its advantages, for example that it makes q spread its mass more and hence it can end up learning a better proposal for importance sampling.

In Line 107, it is claimed that $\log p_\theta(x)$ cannot be calculated explicitly and hence the generative model generalization gap cannot be estimated well. While it cannot be computed exactly, it can be estimated by using importance sampling, which is equivalent to the [importance weighted autoencoder (IWAE)](https://arxiv.org/abs/1509.00519) objective. In light of my previous questions about the amount of p's vs q's contribution to the genralization gap, it might be nice to see this estimation of the generative model generalization gap.

There doesn't seem to be overfitting in Figures 3 and 5c.

What if we train using reverse half-asleep from the start instead of training using ELBO first, and only then fine tuning $\phi$ using reverse half-asleep?

In the paragraph starting in line 228, data from $p_\theta(x)$ is simulated by drawing $z \sim q_\phi(z) = \int \hat p(x) q_\phi(z | x) \mathrm dx$ instead of $z \sim p(z)$ before drawing $x \sim p_\theta(x | z)$. Is this really that different from just drawing from $\hat p(x)$ directly? Especially when looking at Figure 5c, it seems like there is underfitting going on, rather than overfitting. What would have happened if we used $p(z)$ instead of $q_\phi(z)$?

**Strengths And Weaknesses:**

# Originality
As far as I know, this is an original view on the problem and an original solution.

# Quality
I'd like to see some of the following concerns addressed.

I am not entirely convinced that the generalization gap is mainly due to the amortized inference network, as claimed. I understand that in the case of this particular VAE architecture and the MNIST dataset, this is the case, as shown in Figure 2, but what about different model-dataset settings? What if the generative model family is more expressive, e.g. a PixelCNN? Surely some of p must be contributing to the generalization gap there. Another reason to believe this is that for more complex datasets like CIFAR10, there doesn't seem to be overfitting, e.g. as seen in Figure 5c.

The solution to the problem (using reverse half-asleep) only works if there is no generalization gap due to p. If there is a generalization gap due to p, the solution is ineffective. As I understand it, the proposed method is to
1. Train using $E_{\hat p(x)}[ELBO]$ and then
2. Fine tune $\phi$ (parameters of the inference network) using $E_{m(x)}[ELBO]$ where $m(x) = \alpha p_\theta(x) + (1 - \alpha) \hat p(x)$.

As suggested in lines 189-191 ("Intuitively, this is because ..."), this solution is effective because the inference network gets to be trained on a dataset, augmented by samples from the model, $p_\theta(x)$. But if the model $p_\theta(x)$ is trained on $\hat p(x)$, how good can it be? In the extreme case of an infinitely expressive $p_\theta(x)$, it will converge to $\hat p(x)$ and $m(x) = \hat p(x)$, rendering reverse half-asleep to be the same as normal ELBO.

# Clarity
Overall, the paper is clearly written. The compression section (Section 5) is difficult to understand without further context (e.g. what is the stack in Algorithms 1 and 2?), but the high level points are clear. Plot labels would have been useful. Small typos:
- Line 153: Monte Carlo
- Line 240: Equation number missing
- Line 292 Sacrificing

# Significance
Generalization of amortized inference is an important problem.

---

> ### Author Response · Authors · 2022-08-02
> **Thanks and Clarifications, part 2**
>
> **3. Perfect model assumption and  over-fitted model:**
> We want to highlight the perfect model assumption in Section 3 is proposed to show that the classic amortized inference is not consistent whereas wake-sleep/reverse sleep/reverse half-sleep are consistent when we have a perfect model. When we don't have a perfect model,  the reverse half-asleep  $\Big\langle\mathrm{KL}(q\_\phi(z|x)||p\_\theta(z|x)\Big\rangle\_{m(x)}$ where $m(x)=\alpha \hat{p}\_d(x)+ (1-\alpha)p\_\theta$
> can be rewritten (also re-scale by $\frac{1}{\alpha}$) as
> $\Big\langle\mathrm{KL}(q\_\phi(z|x)||p\_\theta(z|x)\Big\rangle\_{\hat{p}(x)}+\frac{1-\alpha}{\alpha}\Big\langle\mathrm{KL}(q\_\phi(z|x)||p\_\theta(z|x)\Big\rangle\_{p\_\theta(x)}$, whose optimum requires $\Big\langle\mathrm{KL}(q\_\phi(z|x)||p\_\theta(z|x)\Big\rangle\_{\hat{p}(x)}=0$ and is the same as the optimal requirement in the classic amortized inference. Therefore, the second term $\frac{1-\alpha}{\alpha}\Big\langle\mathrm{KL}(q\_\phi(z|x)||p\_\theta(z|x)\Big\rangle\_{p\_\theta(x)}$ can be seen as a regularizer on the classic objective. We do agree with the reviewer when $p\_\theta=\hat{p}$, the regularizer is no longer effective but **the reason that this regularizer can alleviate overfitting in practice is that we are using the fact that the model $p\_\theta$ is ``smoother'' than the empirical distribution $\hat{p}$**. This is also the reason why deep learning works.
> For example, if a classifier is learned to be the same as the empirical conditional distribution $\hat{p}(y|x)$, then there is no generalization of that classifier. Therefore, the generalization requires that the model is smooth (i.e. constrained model complexity or hypothesis classes[1]). We have added the relevant discussion in Section 3.3.
>
> **4. Sample from $q\_\phi(z)$:** Since we assume $q\_\theta(z|x)$ is a Gaussian, sampling from $q\_\theta(z|x)$ rather than taking the mean will create a smooth manifold that contains the empirical $\hat{p}$, which we believe is the reason of helping generalization (see the previous point). However, if we sample from $p(z)$, we found that the generated samples are not valid due to the mismatch of $p(z)$ and $q(z)$, which is an independent open problem of the VAE.
>
> **5. One-stage training:** We want to thank the reviewer (also mentioned by ofEq) for this suggestion. In Appendix D of the revised paper, we show that using the proposed Reverse Half-asleep objective from the beginning will lead to a better test BPD comparing to the post-hoc method on the Binary MNIST data. This is consistent with the previous finding in the variational inference literature which suggests a poor variational distribution in the E step may cause the training of the generative model $p_\theta(x|z)$ trap in a local minimum [2]. We hypothesize that when the dataset becomes more complex (like CIFAR10), the poor data sample quality in the beginning of the training can also potentially bias the training and an annealing scheme of the alpha value can also be applied in this case. We will conduct a study in the camera-ready version.
>
>
> **6. Clarity:** We thank the reviewer for pointing out the typos, which we have fixed in the revised paper. We will add an introduction section of the VAE with bits-back coding in the camera-ready version to make the paper more self-contained.
>
> **Reference:**
>
> [1] Shalev-Shwartz S, Ben-David S. Understanding machine learning: From theory to algorithms[M]. Cambridge university press, 2014.
>
> [2] Turner, R, Sahani M. Two problems with variational expectation maximization for
> time-series models.

---

> > ### Author Response · Authors · 2022-08-08
> > **Thanks**
> >
> > We want to thank the reviewer for the valuable advice, which we believe help us greatly improve the paper. We are wondering have our replies addressed your concerns? If there are any additional questions, we can try our best to solve them in the last minute. We're very much looking forward to hearing your further feedback!

---

> > > ### Comment · Reviewer_rcEf · 2022-08-08
> > > **Thanks for the clarifications**
> > >
> > > I will increase my score from to a 6.

---

> ### Author Response · Authors · 2022-08-02
> **Thanks and Clarifications, part 1**
>
> We thank the reviewer for raising valuable points on the marginal likelihood estimation and IWAE. We have considered these questions more deeply and significantly improved the paper as a result. We will first discuss these two points and then try to resolve other concerns.
>
> **1. Marginal likelihood Estimation:** We realize since the ELBO can be written as $\mathrm{ELBO}(x)=\log p\_\theta(x)-\mathrm{KL}(q\_\phi(z|x)||p\_\theta(z|x)$, when we evaluate the ELBO using the optimal inference, the second KL term is approximately 0 and the ELBO is tight to the $\log p\_\theta(x)$. We also verify the tightness using the suggested IWAE and found IWAE and optimal ELBO provide similar values, see Appendix A in the revised paper.
> 2. **KL Directions and IWAE:** Yes, this is a good point. The $\mathrm{KL}(p\_\theta(z|x)||p\_\phi(z|x))$
> used in the Wake-sleep training can encourage the $q$ to match the moment of $p$, which can potentially cover the high-density region of $p$ and improve the importance sampling. However, if the goal is to do compression, minimizing $\mathrm{KL}(p||q)$ doesn't encourage maximizing the ELBO (see Figure 4). An interesting question is if the proposed Reverse Half-asleep training can improve IWAE? Since our method can improve the ELBO without changing the decoder $p_\theta(x|z)$, which means $q\_\phi(z|x)$ will be closer to $p\_\theta(z|x)$ under the $\mathrm{KL}(q\_\phi(z|x)||p\_\theta(z|x))$ divergence measure. In Appendix A, Figure b and c, we show that the posterior learned by our method can improve the IWAE, which indicates the posterior learned by our method can be used as a better proposal distribution. The optimal proposal distribution is obtained when $\mathrm{KL}(q\_\phi(z|x)||p\_\theta(z|x)=0$ ($q\_\phi(z|x)=p\_\theta(z|x)$).
>
>
> **2. Overfitting and Generalization Gap**
> We want to highlight that although the ELBO (which is affected by both encoder and decoder) curve doesn't overfit in Figure 5c, the existence of the amortized inference gap indicates the overfitting of the amortized inference (encoder).
> As we discussed in point 1, the ELBO with optimal inference  can reflect the generative generalization gap up to a constant: $\mathrm{KL}(p\_d(x)||p\_\theta)\approx \frac{1}{M}\sum\_{m=1}^M \log p\_\theta(x_m)+const.$ and higher ELBO (or equivalently smaller BPD) represents less overfitting/better generalization. In Figure 5, we found when the dataset becomes more complex (binary, grey, color), the optimal inference curve becomes less likely to overfit, **which means the generative model doesn't overfit.** However, the gap between the classic ELBO and ELBO with optimal inference still exits, which means the **amortized posterior learned by the classic ELBO overfits to the training data (doesn't generalize well to the test data)**. This gives us room to improve the generalization of the amortized inference. Figure 3 assumes we know the perfect model, we observe the classic VAE training (which corresponds to the wake training) overfits (slightly goes up) but the reverse sleep and reverse half-asleep can consistently improve the BPD, which is consistent with our argument.

---

### Official Review · Reviewer_Ayo3 · 2022-07-11

**Rating:** 7
**Confidence:** 3
**Soundness:** 3 good
**Presentation:** 4 excellent
**Contribution:** 3 good

**Summary:**

The paper studies the generalization performance of Variational Auto-Encoders during test time, measured by the test likelihood. Generative model generalization gap and amortized inference generalization gap are introduced and it is empirically shown that the degradation in generalization is mostly dominated by the inference model. The authors then propose a new variational inference scheme named “reverse half-asleep” which helps close the gap caused by the inference model. Finally, the improvements in generalization and compression rate of using the new scheme over the traditional VAE are demonstrated on Binary MNIST, Grey MNIST, CIFAR10 datasets.

**Questions:**

The improvements might also be shown on CIFAR100 or ImageNet. I just wondered if the authors have performed experiments on those larger datasets. Is the trick for CIFAR10 in line 239 helpful in those settings?

Some minor points:
+ Line 140: missing the Equation number.
+ Line 108: missing the negative sign of the KL estimation.
+ It is clearer to include labels for axes in the figures.

**Limitations:**

Nothing to report.

**Strengths And Weaknesses:**

The paper is very well written. I can understand most of the contents in one read.

The novelty of the paper is the introduction of using reverse KL in the sleep phase during wake-sleep training. It is surprising to me that people have not done it previously.

I think that this is an important paper. The technique is pretty straightforward to implement, hence would be easily adopted in practice.

My main concern, ~which is not addressed in the paper~, is on how using “reverse half-asleep” affects other metrics (like accuracy) during both training and testing phases. **Updated:** The authors pointed out in the response below that the performance on downstream classification tasks can be found in Appendix B.

It is only shown through Figure 2 that the overfitting of the VAE is mainly dominated by the overfitting of the amortized inference network. It would make the paper more technically sound if there is an analytical proof for that observation.

---

> ### Author Response · Authors · 2022-08-02
> **Thanks and clarifications**
>
> We thank the reviewer for the time spent and their valuable feedback that improves the quality of our work.
>
> 1. **Affect other metrics** In our plots in the paper, we show the training BPD  (which corresponds to negative ELBO) and test BPD. In Appendix A and B we add two experiments that demonstrate our method can also improve IWAE and classification accuracy in the down-stream task. We kindly ask the reviewer to expand on what other metrics they had in mind.
>
>
> 2. **Theoretical Proof** Yes we agree with the reviewer it would be good to have some theoretical proof of the overfitting phenomenon. However, it is not easy, we didn't realize there exists any generalization bounds on the generalization model in terms of the test likelihood (or ELBO).  However, we notice that the proposed reverse half-asleep can be seen as a regularizer adding to the classic inference objective (see Section 3.3 in the revised paper). This regularization view may explain why it can help alleviate overfitting: it penalizes the hypothesis space of the amortized network. We leave the analysis of how the regularizer affect the generalization into future work.
>
> 3. **Large Images dataset:** We have done an additional experiment on CIFAR100. We train a VAE with latent dimension 256 for 500 epochs on CIFAR100, the VAE architecture and training schemes are the same as that used for CIFAR 10 (Section 4). After training, we apply the proposed reverse half-asleep method (with the sample from $q(z)$) to train the encoder with  200 epochs with a learning rate $1e^{-5}$ and Adam optimizer. We found the BPD improved from 5.278 to 5.267 (lower is better), which indicates our method is helpful in this setting.
>
>
> 4. **Typos:** We thank the reviewer for pointing out the typos, we have fixed that in the revised version.

---

> > ### Comment · Reviewer_Ayo3 · 2022-08-07
> > **Response to authors**
> >
> > Thank you for your clarification! I'm happy to increase my rating after re-reading the paper.
> >
> > Regarding to point 3, **large Images dataset**, because the trick used in the experiments is important, I think it is better to mention it in Section 3.2, rather than in the experimental section.

---

> > > ### Author Response · Authors · 2022-08-08
> > > **Thanks again!**
> > >
> > > We thank the reviewer again for raising the score and the additional advice, we will improve the structure of the paper based on the advice in the camera-ready version.

---

### Official Review · Reviewer_syyo · 2022-07-12

**Rating:** 6
**Confidence:** 4
**Soundness:** 4 excellent
**Presentation:** 4 excellent
**Contribution:** 3 good

**Summary:**

The authors investigate the generalization of (amortized) variational auto-encoders in terms of their test likelihood. It is noted that "overfitting" in VAEs is largely contributed by the amortized inference network. To mitigate such overfitting, the authors develop reverse sleep and reverse half-sleep methods inspired by the wake-sleep algorithm, leading to better compression of the data distribution.

**Questions:**

My comments here are mostly a follow up to the main weakness previously mentioned.

* The authors interestingly note that standard VAE training with ELBO (i.e. the wake phase) only has access to a finite training dataset. The sleep phase allows for virtually unlimited training data, and hence mitigates overfitting. One can subsequently think of how does the applied fix to the amortized network behave for the following applications other than compression to name a few:
  * out-of-distribution detection/generalization
  * sample generation
  * semi-supervised learning

* Another question to explore would be does this make VAEs better utilize the latent space? From the compression results, perhaps the answer is promising and tending towards yes. Is it more robust to the choice of the dimensionality of the latent space as a consequence of improving the amortization gap?

I am curious if the authors have a plan for this, or could provide a small proof-of-concept for something beyond compression.

**Limitations:**

Yes.

**Strengths And Weaknesses:**

Strengths:

* The authors develop on a likelihood-based characterization of what it means for VAEs to overfit. The demonstration on a toy example which clearly demonstrates the overfitting phenomenon is much appreciated.

* The step-by-step development of the method with supporting experiments provides a relatively clear picture.

Weaknesses:

It would appear that the method specifically caters to optimizing for best compression performance. Even in their own experiments, the authors note that without the half-sleep correction in the sleep phase, the generated samples may be degraded and not representative of the true data distribution.

This makes me wonder that the notion of test likelihood for generalization remains very limited in scope, and in principle only caters to compression performance (this probably would not be an issue in the case of a well-specified model, which is virtually impossible to achieve). In that sense, the insights developed in this paper may not hold across broader family of generative models.

In light of such missing broader evaluation of direct implications, I will be keeping my score to a weak accept.

Minor:

* Is there a typo in the summands in Eq. (8) where you need expectation w.r.t $q^*(z \mid x^\prime_m)$ and $q_{\hat{\phi}}(z \mid x^\prime_m)$ instead of just conditioning on $x$?

* Missing closing ")" in Eq. (5).

* Line 153: Typo in "Mon-Carlo".

---

> ### Author Response · Authors · 2022-08-02
> **Thanks and Clarifications**
>
> We thank the reviewer for the time spent and their valuable suggestions, we have added the new experiments in the revised paper.
>
> **1. Other applications:** We thank the reviewer for the valuable feedback, which has encouraged us to find other possibilities for our method. We noticed that since our method can improve the ELBO without changing the decoder since the ELBO can be rewritten as $\mathrm{ELBO}(x)=\log p\_\theta(x)-\mathrm{KL}(q\_\phi(z|x)||p\_\theta(z|x)$, higher ELBO value means the approximated posterior is closer to the true posterior under the KL measure! Therefore, the learned posterior can definitely benefit for other applications. In Appendix A (Figures b and c), we show that the learned proposed posterior can be used as a better proposal distribution to improve the importance weighted auto-encoder. In Appendix B, we apply the learned posterior to generate representations for the down-stream classification task, which can consistently achieve better performance compared to the classic VAE. Similarly, this means it could also improve semi-supervised learning which will also utilize the latent space.  Our method is proposed to improve the in-distribution generalization, which in principle will benefit likelihood-based OOD detection since it will assign higher density to the in-distribution data. Paper [1] shows that OOD detection and generalization are two counter problems, encouraging OOD generalization will degrade OOD detection, so it may not help the OOD generalization. We leave a detailed study to future work.
>
> **2. Latent dimension:** In Appendix C of the revised paper, we train a VAE with different latent dimensions [16,64,128] on the Binary MNIST dataset and find overfitting happens in all cases. We also apply the proposed method in these settings and find our method can also consistently improve the performance.
>
> **3. Others:** We are grateful for pointing out the typos, we have fixed that in the revised version.

---

> > ### Comment · Reviewer_syyo · 2022-08-03
> > **Response to Author Rebuttal**
> >
> > I appreciate the authors taking the time to test of these proof-of-concept experiments, where clearly it can lead to promising future research.
> >
> > I still vote for acceptance but keep my score the same to reflect the degree of confidence in those future directions. Thank you!

---

> > > ### Author Response · Authors · 2022-08-08
> > > **Thank you!**
> > >
> > >  We want to thank the reviewer again for their time and valuable suggestions! We are encouraged by the acknowledgments from the reviewer.

---

### Official Review · Reviewer_ofFq · 2022-07-13

**Rating:** 6
**Confidence:** 3
**Soundness:** 3 good
**Presentation:** 3 good
**Contribution:** 3 good

**Summary:**

The authors examine the amortization inference generalization gap with the motivation of improving performance on lossless compresseion and image modelling. They introduce the reverse sleep amortized inference and reverse half-asleep amortized inference, components of a modified wake-sleep algorithm, and use these algorithms to train a VAE for MNIST, CIFAR10 and lossless compression experiments.

**Questions:**

How does a VAE perform when trained in one step (all of the parameters) using the half-asleep objective, ie. adding the additional expectation to the standard ELBO training, as oppose to the two-stage wake-sleep method?

The test error in figure 5 is still increasing with each epoch. Is this still an indication of overfitting?

On line 239 you state that you sample from q(z) instead of p(z). How is this done? Do you Monte Carlo marginalize out the training data from q(z|x)p(x)?

Typos:
Line 117: "the VAE that is described in Figure 2". No description is given here.
Equation 12: The expectation is written over the joint p(x,y) but I think you want p(x).
Line 240: Equation ?.

**Limitations:**

No discussion of the limitations of the method was present.

**Strengths And Weaknesses:**

Strengths:

The investigation is well motivated by the application to lossless compression.
I thought that the intuition presented for the aspects of the lack of generalization in section 2 were good.

Weaknesses:
I did not think the intuitions in section 2 were well supported by they toy experiments presented.

I did not find the exposition clear. I often found myself confused when the authors explained background that I am already familiar with. The authors would introduce a rearrangement of the ELBO, as in equation 2, only to immediately introduce another rearrangement in equation 3 leaving me wondering why early equations were introduced. I did not find the explanation of the variational family starting on line 56 very clear.

The method used in the paper is very similar to the one used in [4] Cemgil et al. 2020, as noted in the related work. Unfortunately, there are not a lot of additional contributions to distinguish this paper from [4], other than a reinterpretation of the method.

The authors presented a very long explanation for what is a relatively simple idea: they want the inference network to converge to the correct posterior if they model converges. This does not happen in the standard ELBO because the expectation is only over the training data. They authors bootstrap their own model distribution by effectively adding an expectation over model simulated data to the ELBO. Under the very strong assumption that the model converges to the true distribution, you will me minimizing the KL between all true posteriors and q (which isn't amortization anymore), and with a flexible enough q parametrization, you will satisfy your requirement.

The model converging to the true distribution is a very strong assumption. Not much investigation is presented for whether this is a reasonable assumption (other than the fact that the inference network dominated the error in a toy example) or issues that occur when this assumption doesn't hold, which would be almost all of the time. Limitations are not discussed. A modification (half-asleep) is presented to address this issue.

The experimental section is not very strong, only applying the method to MNIST, CIFAR10 and a single compression experiment. Standard regularization techniques were not compared against. In fact, no other training methods other than those introduced in the paper were examined, including IWAE mentioned in the related work.

I think it's a good idea to examine this particular aspect of VAE overfiiting but I recommend major revisions before conference acceptance.

---

> ### Author Response · Authors · 2022-08-02
> **Thanks and Clarifications part 2**
>
> **5. Regularization comparison and IWAE:** In Section 4 (Table 1 and Figure 6), we have compared our method with a regularizer that is specially designed to alleviate the overfitting of the amortized inference from [3].  We showed that our method can **consistently outperform the regularization method**. We hypothesize that it is because the regularization method is blind to the data manifold whereas our method explicitly utilizes the data manifold learned by the decoder to `regularize' the training of the amortized inference. We thank the reviewer for mentioning IWAE and in the revised paper (Appendix A), we add an IWAE experiment which shows that **the posterior learned by the proposed method can be served as a better proposal distribution and thus improve the IWAE performance**.
>
>
> **6. Sample from $q\_\phi(z)$:** Sampling from $q\_\phi(z)$ is conducted by ancestor sampling where we first sample $x'\sim p_d(x)$ (from the training dataset) and then sample $z'\sim q\_\phi(z|x=x')$.
>
> **7. One-step training:**  We thank the reviewer (also mentioned by rcEf) for the great suggestion. In Appendix D, we show that using the proposed Reverse Half-asleep objective from the beginning will lead to a better test BPD compared to the post-hoc method. This is consistent with the previous finding in the variational inference literature which suggests a poor variational distribution in the E step may cause the training of the generative model $p_\theta(x|z)$ to become trapped in local minima [4]. We hypothesize that when the dataset becomes more complex (like CIFAR10), the poor data sample quality in the beginning of the training can also potentially bias the training, and an annealing scheme for the alpha value can also be applied in this case. We will conduct a study in the camera-ready version.
>
> **8. The test error increases:** Yes, in both Figure 2 and 5a, we found the test error for the optimal inference increase, **which is caused by the over-fitting of the decoder**, and relates to the generative model generation gap. We also find when the dataset becomes more complex (Grey MNIST in Figure 5b and CIFAR in Figure 5c), the over-fitting of the decoder no longer happens. However, our paper focuses on improving the generalization of amortized inference. The decoder in all the Figures is shared and fixed, so a better ELBO indicates better generalization of the amortized posterior. We leave the study of the generalization of the decoder to future work.
>
> **9. Typos:** We thank the reviewer for pointing out the typos, which we have fixed in the revised paper. The expectation in Equation 12 (Equation 13 in the revised paper) is over x since the integration over z is included in the ELBO (Equation 1).
>
> ### Reference:
> [1] Cemgil T, Ghaisas S, Dvijotham K, et al. The autoencoding variational autoencoder. Advances in Neural Information Processing Systems, 2020
>
> [2] Chen T, Kornblith S, Norouzi M, et al. A simple framework for contrastive learning of visual representations. International conference on machine learning, 2020
>
> [3] Shu R, Bui H H, Zhao S, et al. Amortized inference regularization. Advances in Neural Information Processing Systems, 2018.
>
> [4] Turner, R, Sahani M. Two problems with variational expectation maximisation for
> time-series models.

---

> > ### Author Response · Authors · 2022-08-08
> > **Thanks for the reviews**
> >
> > We want to thank the reviewer for the comments and questions, which definitely help greatly improve the paper.
> > We are wondering have our replies addressed your concerns or make you further change the score of the paper? If there are any additional questions, we can try our best to solve them in the last minute. We're very much looking forward to hearing your further feedback on the response.

---

> > > ### Comment · Reviewer_ofFq · 2022-08-09
> > > **Re: Thanks for the reviews**
> > >
> > > Thanks to the authors for the in-depth response. Your response and modifications to the paper have addressed the majority of my concerns. In particular I'm glad that you have added an IWAE and a one-step experiment justifying the use of your particular method. I still would have liked to see more challenging benchmarks than MNIST and CIFAR10 in your experiment section but I am comfortable increasing my score after your improvements.

---

> > > > ### Author Response · Authors · 2022-08-09
> > > > **Thanks!**
> > > >
> > > > We want to thank the for the feedback and the acknowledgement. In the reply to reviewer Ayo3, we have done an additional experiment that shows our method can also improve cifar100, we will add that to the revised paper. Thanks!

---

> ### Author Response · Authors · 2022-08-02
> **Thanks and Clarifications, part 1**
>
> **1. Toy experiment in Section 2:** We thank the reviewer (also reviewer ecEf) for raising the question about the toy problem, which lead us to think further on this toy example and come up a new demonstration in the revised paper. The motivation of the toy problem is to visualize both the generative model generation gap $\mathrm{KL}(p\_d(x)||p\_\theta(x))\approx \frac{1}{M}\sum\_{m=1}^M \log p\_\theta(x\_m')+const.$ and the amortized inference generalization gap. However, the evaluation of $\log p\_\theta(x)$ is intractable and we can only access the ELBO, but we realize that when use the optimal inference strategy to get $q^{\*}(z|x\_m')$, the ELBO is actually tight to the likelihood since $\mathrm{ELBO}(x)=\log p\_\theta(x)-\mathrm{KL}(q\_\phi(z|x)||p\_\theta(z|x)$ and the second KL term is approximately zero for the $q^{*}(z|x)$.
> We also verify the tightness using IWAE (see Appendix A in the revised paper). Therefore, **the optimal inference ELBO can accurately reflect the generative model generation gap up to a constant** (a larger negative ELBO indicates a larger gap). At the same time, the gap between optimal inference ELBO and the classic amortized inference ELBO visualizes the amortized inference generalization gap. We can see during training, that the generative model generalization gap has a negligible increase compared to the amortized inference gap, which supports the intuition that the overfitting, in this case, is dominated by the overfitting of the amortized inference.
>
> **2. Clarity of the VAE introduction:** We thank the reviewer for the advice on clarity, we have re-written the corresponding content to improve the demonstration of the introduction section, see Section 2 in the revised paper.
>
> **3. Relation to [1]:**
> We thank the reviewer for pointing this out and we realize the discussion about paper [1] in our related work section is not sufficient which may cause confusion. We want to clarify that **both the methodology and goal in paper [1] are completely different from ours.**
> Intuitively, their paper considers the following data generation process $x\_1 \sim p\_d(x)$, $z\_1 \sim q(z|x=x\_1)$, $x\_2 \sim p(x|z=z\_1)$ and propose to enforce the latent consistency between $q(z|x\_1)$ and $q(z|x\_2)$ for paired data $(x\_1,x\_2)$ to encourage the robustness of the learned representation (see equation 10 in [1]). This procedure is close to the self-supervised contrasting learning method [2] where the augmented data is the reconstruction of the training data using the VAE model.
> In our paper, we want to encourage the sample from the model $x'\sim \int p_\theta(x|z)p(z)dz$ to have high ELBO under the model to improve the generalization of the amortized inference, and **no paired data is required in our procedure**. We have made this clear in the revised paper (see Appendix D) and hope it can resolve the reviewer’s concern.
>
> **4. Assumption of perfect model:** We want to highlight that the generalization of the amortized inference is similar to the generalization of the classification problem, we cannot guarantee to recover the true distribution given only finite data. The perfect model assumption is proposed to demonstrate the classic amortized inference training objective doesn't satisfy the consistency requirement but wake-sleep/reverse sleep/reverse half-sleep are consistent for a perfect model. When the model is not perfect,
>  the proposed reverse half sleep criterion $\Big\langle\mathrm{KL}(q\_\phi(z|x)||p\_\theta(z|x)\Big\rangle\_{m(x)}$ where $m(x)=\alpha \hat{p}\_d(x)+ (1-\alpha)p_\theta(x)$
> can be rewritten (also re-scale by $\frac{1}{\alpha}$) as
> $\Big\langle\mathrm{KL}(q\_\phi(z|x)||p\_\theta(z|x)\Big\rangle\_{\hat{p}(x)}+\frac{1-\alpha}{\alpha}\Big\langle\mathrm{KL}(q\_\phi(z|x)||p\_\theta(z|x)\Big\rangle\_{p\_\theta(x)}$ whose optimum requires $\Big\langle\mathrm{KL}(q\_\phi(z|x)||p\_\theta(z|x)\Big\rangle\_{\hat{p}(x)}=0$, which is the same training requirement as the classic amortized inference. Therefore, **when the model is perfect, our method can be seen as a regularizer to penalize the hypothesis space of the amortized network to alleviate over-fitting**. We have added the relevant discussion in Section 3.3 and we want to thank the reviewer for raising this concern.

---

### Author Response · Authors · 2022-08-02
**Updates on the revised paper**

We thank the reviewer for their time and valuable suggestions for the paper, we truly believe these suggestions significantly improve the paper's quality. Based on the suggestions, we have  modified the paper and appendix with the following changes:
1. In Section 2, we provide a new interpretation of the toy problem with an empirical proof of the tightness of the bound in Appendix A.
2. In Section 3.1, we discuss that when the model is not perfect, our method can be seen as a regularizer.
3. In Appendix A, we show that the posterior learned by our method can improve the IWAE results.
4. In Appendix B, we show that our method can benefit the downstream classification task.
5. In Appendix C, we show that the overfitting phenomenon still exists when using a larger latent dimension.
6. In Appendix D, we apply the proposed method to train the model from the beginning, and found it still improves the test ELBO.
7. In Appendix E, we add a detailed discussion about the paper [1] to show their methods and motivations are completely different from ours.
8. We temporarily move the related work to Appendix D due to the page limit and the short rebuttal time, we will reform the structure of the paper in the camera-ready version.

**Reference**

[1] A. T. Cemgil, S. Ghaisas, K. Dvijotham, S. Gowal, and P. Kohli. Autoencoding variational
autoencoder. Neural Information Processing Systems, 2020

---

### Meta-Review · Area_Chair_UFvX · 2022-09-04

**Recommendation:** Accept
**Confidence:** Less certain

**Metareview:**

The paper studies generalization in generative models and highlights two factors for the lack of generalization, generalization gap of the model itself and generalization gap in the ELBO objective caused due to amortized inference. The authors experimentally show that the latter is the more prevalent reason. To alleviate that the authors propose a reverse wake sleep algorithm which is based on training from a mixture of samples from the true distribution and the model's distribution. This latter algorithm is the primary contribution of the paper which is a novel algorithm as recognized by the reviewers. The reviewers have found the experiments compelling and the conceptual contribution significant but easy to implement thus projecting general adoption. Reviewers unanimously agreed upon the contribution of the paper being above the bar for Neurips.

**Award:**

No

---

### Decision · Program_Chairs · 2022-09-14

Accept